# Prognostic Value of Sarcopenia and Metabolic Parameters of ^18^F-FDG-PET/CT in Patients with Advanced Gastroesophageal Cancer

**DOI:** 10.3390/diagnostics13050838

**Published:** 2023-02-22

**Authors:** Ricarda Hinzpeter, Seyed Ali Mirshahvalad, Roshini Kulanthaivelu, Vanessa Murad, Claudia Ortega, Ur Metser, Zhihui Amy Liu, Elena Elimova, Rebecca K. S. Wong, Jonathan Yeung, Raymond W. Jang, Patrick Veit-Haibach

**Affiliations:** 1Joint Department of Medical Imaging, University Health Network, Sinai Health System, Women’s College Hospital, University of Toronto, Toronto, ON M5G 2C4, Canada; 2Department of Biostatistics, Princess Margaret Cancer Centre, University Health Network, University of Toronto, Toronto, ON M5G 2C4, Canada; 3Department of Medical Oncology, Princess Margaret Cancer Centre, University Health Network, University of Toronto, Toronto, ON M5G 2C4, Canada; 4Department of Radiation Oncology, Princess Margaret Cancer Centre, University Health Network, University of Toronto, Toronto, ON M5G 2C4, Canada; 5Division of Thoracic Surgery, Department of Surgery, Toronto General Hospital, University Health Network, University of Toronto, Toronto, ON M5G 2C4, Canada

**Keywords:** ^18^F-FDG PET/CT, sarcopenia, gastroesophageal cancer

## Abstract

We investigated the prognostic value of sarcopenia measurements and metabolic parameters of primary tumors derived from ^18^F-FDG-PET/CT among patients with primary, metastatic esophageal and gastroesophageal cancer. A total of 128 patients (26 females; 102 males; mean age 63.5 ± 11.7 years; age range: 29–91 years) with advanced metastatic gastroesophageal cancer who underwent ^18^F-FDG-PET/CT as part of their initial staging between November 2008 and December 2019 were included. Mean and maximum standardized uptake value (SUV) and SUV normalized by lean body mass (SUL) were measured. Skeletal muscle index (SMI) was measured at the level of L3 on the CT component of the ^18^F-FDG-PET/CT. Sarcopenia was defined as SMI < 34.4 cm^2^/m^2^ in women and <45.4 cm^2^/m^2^ in men. A total of 60/128 patients (47%) had sarcopenia on baseline ^18^F-FDG-PET/CT. Mean SMI in patients with sarcopenia was 29.7 cm^2^/m^2^ in females and 37.5 cm^2^/m^2^ in males. In a univariable analysis, ECOG (<0.001), bone metastases (*p* = 0.028), SMI (*p* = 0.0075) and dichotomized sarcopenia score (*p* = 0.033) were significant prognostic factors for overall survival (OS) and progression-free survival (PFS). Age was a poor prognostic factor for OS (*p* = 0.017). Standard metabolic parameters were not statistically significant in the univariable analysis and thus were not evaluated further. In a multivariable analysis, ECOG (*p* < 0.001) and bone metastases (*p* = 0.019) remained significant poor prognostic factors for OS and PFS. The final model demonstrated improved OS and PFS prognostication when combining clinical parameters with imaging-derived sarcopenia measurements but not metabolic tumor parameters. In summary, the combination of clinical parameters and sarcopenia status, but not standard metabolic values from ^18^F-FDG-PET/CT, may improve survival prognostication in patients with advanced, metastatic gastroesophageal cancer.

## 1. Introduction

Esophageal, gastroesophageal and gastric cancers are major causes of cancer-associated morbidity and death worldwide [1]. Despite the ongoing development of novel therapeutic strategies, the prognosis of these entities remains poor, with a 5-year survival rate between 5–46% [2]. In addition, up to 50% of all patients are diagnosed with advanced stage of disease at the time of initial presentation, precluding curative treatment [3].

Fluorine-18-Fluorodeoxyglucose positron emission tomography/computed tomography (^18^F-FDG-PET/CT) is an established and important tool in the workup of esophageal, gastroesophageal and gastric cancers, providing significant diagnostic and prognostic value amongst these patients [4,5]. Additionally, the CT component allows for the assessment of skeletal muscle and sarcopenia.

Skeletal muscle depletion, also known as sarcopenia, is the involuntary loss of muscle mass. This is one of the main components of cancer cachexia syndrome, which is associated with mobility disorder, loss of independence and even increased risk of death [6]. Prior studies have emphasized the influence of nutritional state and body composition on overall survival in various tumor entities [7,8]. Significant weight loss due to dysphagia and altered eating habits is a well-documented clinical problem in patients with gastroesophageal cancers, with a prevalence of up to 79% prior to surgery [8,9,10,11]. Although evidence has shown a significant correlation between sarcopenia and major postoperative complications, the prognostic value of sarcopenia has not been definitively established in patients with advanced disease [12,13].

As a result, it is highly desirable to further investigate potential prognostic factors which could support therapeutic decision making in patients with esophageal and gastroesophageal cancers. Therefore, the aim of our study was to determine the prognostic value of sarcopenia measurements and metabolic activity parameters of primary gastroesophageal cancer in patients with advanced metastatic disease.

## 2. Materials and Methods

Between November 2008 and December 2019, 128 patients with primary metastatic esophageal or gastroesophageal cancer who underwent ^18^F-FDG-PET/CT as part of their initial staging were included from an institutional registry. Patients with primary metastatic disease who were missing staging 18F-FDG-PET/CT (*n* = 35) were excluded from the study. Demographic data of the study cohort are provided in Table 1. Different aspects about sarcopenia and PET/CT radiomics in patients with gastroesophageal cancer from the same patient population were evaluated in a different manuscript [14]. 

This retrospective study was approved by the institutional review board and the need to obtain informed consent from patients was waived (REB# 19-5575). 

### 2.1. Imaging Acquisition

Whole body ^18^F-FDG PET/CT was acquired prior to treatment on a Siemens mCT40 (Siemens Healthineers, Erlangen, Germany). Images were obtained from the skull base to the upper thighs. Iodinated oral contrast media was administered for bowel opacification; no intravenous contrast media were used. Patients received 300–400 Mbq (4–5 MBq/kg) of ^18^F-Fluorodeoxyglucose (FDG) after having fasted for 6 hours, and PET/CT image acquisition was performed after approximately 60 min. Overall, 5–9 bed positions were obtained, depending on patient height, with an acquisition time of 2–3 min per bed position. CT parameters were 120 kVp tube voltage, 3.0 mm slice width, 2 mm collimation, 0.8 s rotation time and 8.4 mm feed/rotation.

### 2.2. Image Analysis and Sarcopenia Measurements

The mean, max, and peak standardized uptake values (SUV) and SUV normalized by lean body mass (SUL), were collected from the primary tumor in each patient, using a common commercially available imaging software (Mirada XD Workstation, Mirada Medical, Ltd., Oxford, UK). SUV were obtained manually with a volume-of-interest (VOI) covering the entire tumor volume as defined by PET. Sarcopenia measurements were taken from the CT component of the ^18^F-FDG PET/CT. Assessment of skeletal muscle mass was performed at the level of the third lumbar vertebra using Slice-O-Matic (TomoVision, version 5.0, Magog, QC, Canada) Hounsfield units (HU) were used to identify skeletal muscle (threshold −29 to 150 HU) (Figure 1). Skeletal muscle index (SMI) was calculated by normalizing the muscle area (cm^2^) for the subject’s height in squared meters (m^2^). SMI cutoff values for sarcopenia were used as follows [15]: SMI of 34.4 cm^2^/m^2^ in females and SMI of 45.4 cm^2^/m^2^ in males. Image analysis was performed by one radiologist with 5 years of experience in oncologic imaging.

### 2.3. Statistical Analysis

Summary statistics were used to describe demographics and disease characteristics. Kaplan–Meier (KM) method was used to estimate overall survival (OS) and progression-free survival (PFS) and there 95% confidence intervals (CI). Univariable analysis (UVA) was used to identify potential prognostic factors for OS and PFS, including clinical variables, SUV parameters and anthropometric indices. Parameters with a *p*-value of <0.05 were included in a subsequent analysis to build a multivariable analysis (MVA). Model performance was quantified and visualized using area under the time-dependent receiver operating characteristic (ROC) curve (AUC), and calculated using leave-one-out cross-validation which served as an internal validation method. All statistical analyses were carried out in R version 4.0.2 [16] and a *p*-value of <0.05 was considered statistically significant.

## 3. Results

### 3.1. Baseline Characteristics of the Study Cohort

Overall, 128 patients (26 females, 102 males; mean age 64 ± 11 years, range: 29–91 years) with advanced metastatic gastroesophageal squamous cell carcinoma (*n* = 44) and adenocarcinoma (*n* = 84) were included in this study. The majority of patients had an ECOG score of 0 or 1 (22% and 57%, respectively) and 21% had an ECOG score of 2 or above.

All patients were deemed palliative and underwent either chemotherapy or radiotherapy or a combination of both. A total of 2/128 patients underwent additional salvage esophagectomy and esophago-gastrectomy.

At the time of diagnosis, 117/128 (91%) patients presented with regional lymph node metastases and concurrent distant metastatic disease to extra-regional lymph nodes, liver, bone, brain or peritoneum. In addition, 6/128 cases had distant metastases only and in 5/128 cases the N-stage was undetermined (Table 1).

### 3.2. Image Analysis and Sarcopenia Measurements

All primary tumors were associated with increased metabolic activity on staging ^18^F-FDG-PET/CT with a mean SUVmax of 15.4, ranging from 4.1 to 54.4. Further SUV parameters are summarized in Table 2.

Overall, 60/128 (47%) patients had an SMI score below the cutoff value for sarcopenia, indicating low skeletal muscle mass and poor nutritional status. The mean SMI score in patients with sarcopenia was 29.7 cm^2^/m^2^ in females and 37.5 cm^2^/m^2^ in males.

### 3.3. Analysis on Survival Prognostication

The median (95% confidence interval) OS and PFS in our cohort was 9.0 (6.9, 10.7) months and 6.0 (4.7, 7.0) months, respectively. OS and PFS showed statistically significant differences with regard to sarcopenia status. Median OS was 9.9 (7.8, 12.4) months in non-sarcopenic patients and 6.8 (4.9, 10.1) months in patients with sarcopenia (*p* = 0.032). Median PFS was 7.1 (4.6, 9.2) months in non-sarcopenic patients and 5.1 (4.5, 6.8) months in patients with sarcopenia (*p* = 0.02). Statistical analysis did not show significant differences when comparing patients with squamous cell carcinoma and adenocarcinoma regarding OS and PFS (*p* = 0.67 and 0.68, respectively). Consequently, further statistical analysis was performed on the entire cohort.

UVA using Cox proportional hazards revealed the following parameters as poor prognostic factors for OS and PFS: ECOG performance status (*p* < 0.001), bone metastases (*p* = 0.028) and sarcopenic status (dichotomized sarcopenia score (*p* = 0.033) and SMI (0.0075)). Additionally, age was associated with decreased OS in the overall cohort (*p* = 0.017). Metabolic parameters derived from baseline ^18^F-FDG-PET/CT, however, were not significantly associated with decreases in OS and PFS (Table 3).

On MVA, ECOG performance status (*p* < 0.001) and bone metastases (*p* = 0.01 for OS and 0.019 for PFS) remained significant poor prognostic factors for OS and PFS in the overall cohort (Table 4). To this clinical model, we added the sarcopenia status of the patient determined by the SMI score (*p* = 0.065 for OS and 0.03 for PFS). The combined model (clinical parameters + sarcopenia status) outperformed the model with solely clinical parameters over a clinical course of 33 months, indicating improved OS and PFS prognostication when taking into account the patients’ nutritional status. The results were an OS AUC of 0.76, 0.71 and 0.84 for the combined model compared to 0.7, 0.67 and 0.82 for the clinical model at 6, 12 and 33 months of follow-ups, respectively (Figure 2), and PFS AUC of 0.67, 0.69 and 0.83 for the combined model compared to 0.63; 0.65 and 0.7 for the clinical model at 6, 12 and 33 months of follow-ups, respectively (Figure 3).

## 4. Discussion

In our study, we assessed the prognostic value of sarcopenia—an indication for poor nutritional state—in combination with clinical variables and metabolic parameters, derived from ^18^F-FDG-PET/CTs among patients with advanced metastatic esophageal and gastroesophageal cancers. The main finding of our study was that sarcopenia (low SMI value) is a prognostic marker for poor OS and PFS. Furthermore, improved prognostication of OS and PFS was observed when sarcopenia status was combined with clinical variables as opposed to clinical variables only. However, standard metabolic parameters from ^18^F-FDG-PET/CTs obtained from the primary tumor were not associated with an overall improvement in outcome prediction.

Sarcopenia describes a progressive and generalized loss of skeletal muscle mass and function, which is associated with an increase in adverse outcomes, including a high risk of falls, frailty and mortality [17]. The impact of sarcopenia in cancer patients has been studied across a broad range of malignancies and has been shown to be an independent poor prognostic factor among both patients deemed curative and those undergoing palliative treatment [18,19,20]. A recent study by Gu et al. [21] indicates prognostic significance of combined pretreatment body mass index (BMI) and BMI loss in patients with esophageal cancer. However, patients’ BMIs were not found to be significantly different in sarcopenic versus non-sarcopenic patients; neither was it associated with overall survival. This emphasizes the need for advanced screening measurements, besides height and weight, especially since sarcopenia in obese patients is a known phenomenon [18,22].

The impact of sarcopenia in gastroesophageal cancer has been the subject of several previous studies [13,23,24,25,26,27]. A recent study by Sato et al. [25] showed significantly worse overall survival rates in a cohort of 48 patients with locally advanced esophageal squamous cell carcinoma who underwent definite chemoradiotherapy—with a 3-year survival rate of 36.95 % vs. 63.9%. Similarly, Koch et al. [24] investigated the impact of sarcopenia as a prognostic factor for survival in a cohort of 83 patients with locally advanced non-metastatic gastric or gastroesophageal junction (GEJ) cancer, who underwent curative treatment with perioperative chemotherapy and surgery. The authors reported a significantly shorter median survival in patients with sarcopenia compared to non-sarcopenic patients (35 vs. 52 months). Further, perioperative complications occurred more frequently in sarcopenic patients. This is in line with the results of our study, showing a significant decrease in OS (6.8 vs. 9.9, *p* = 0.032) and PFS (5.1 vs. 7.1 months, *p* = 0.02) in gastroesophageal cancer patients with primary palliative treatment intent when sarcopenia is present. Notably, sarcopenic patients in the present study showed lower median OS and PFS compared to the prior studies [24,25], which is likely related to the presence of distant metastasis in our cohort. Additionally, our study proposes the combination of standard clinical parameters with imaging-derived sarcopenia measurements to enhance outcome predictions in these patients over a clinical course of 33 months for OS (AUC 0.7 vs. 076 for 6 months; 0.67 vs. 0.71 for 12 months and 0.82 vs. 0.84 for 33 months) and PFS (AUC 0.63 vs. 0.67 for 6 months; 0.65 vs. 0.69 for 6 months and 0.7 vs. 0.83 for 33 months).

Only a few studies so far have reported contrasting study results [28,29,30], including a study by Grotenhuis et al. [31], who investigated 120 patients undergoing esophagectomies following neoadjuvant chemoradiotherapy for primary esophageal cancer. The results of their study indicate that the presence of sarcopenia is not associated with negative short- and long-term outcomes. Although these studies applied similar measurement techniques for the assessment of sarcopenia (using CT images at the level of the third lumbar vertebra), cutoff SMI values varied between publications. Applying different threshold values for the assessment of sarcopenia is part of an ongoing debate. Whereas some authors used self-developed software tools, recent studies have performed their measurements with frequently used commercially available software, at least partly minimizing the effect of different evaluation approaches. The cutoff values used in the present study are one of the most frequently used within the literature. Further, none of these studies reported the presence of distant metastatic disease in their patient cohort, which may indicate that sarcopenia plays an even more prominent role in outcome prognostication, particularly in advanced metastatic disease.

Although dual X-ray absorptiometry (DXA), magnetic resonance imaging (MRI), computed tomography (CT) and ultrasound (US) have previously been investigated for imaging assessment of sarcopenia, MRI and CT are considered the most suitable methods for analyzing quantitative and qualitative changes in body composition [30]. One reason for that might be also that CT and MRI are the most frequently used cross-sectional imaging methods in cancer patients, and thus availability of sarcopenia measurements from this standard-of-care imaging is certainly higher than compared with the other methods.

^18^F-FDG-PET/CT is an established and routinely used imaging technique for the staging of several different malignancies, including gastroesophageal cancer. ^18^F-FDG-PET/CT has resulted in a significant improvement in imaging assessment and management of gastroesophageal cancer patients at initial staging, treatment planning, restaging as well as response assessment [32]. In the present study, ^18^F-FDG-PET/CT was routinely performed to stage patients with esophageal and gastroesophageal cancer. Assessment of sarcopenia was performed on the CT component of this study. Therefore, in the future, this could potentially provide patients with the one-stop shop imaging-derived means to predict OS and PFS as part of routine clinical management. A study by Mallet et al. [33] had a similar approach, using staging ^18^F-FDG-PET/CT for the assessment of sarcopenia in patients with locally advanced esophageal cancer treated with chemoradiation. This is in line with the results of our study, indicating poor prognosis in sarcopenic patients. However, the analysis in our study was performed on a larger sample size than compared to the aforementioned study. Further, we investigated a more homogenous set of patients by only including those with metastatic esophageal and gastroesophageal cancer—adding potential value to the literature, particularly in patients with advanced disease.

Notably, it would be highly desirable to obtain clinical, nutritional as well as functional imaging data simultaneously; however, as the results of our study demonstrated, adding standard metabolic parameters to the model with clinical information and sarcopenia measurements does not improve the prediction of OS and PFS. In contradiction, several prior studies demonstrated that metabolic parameters of ^18^F-FDG-PET/CT do improve overall survival prediction in patients with gastroesophageal cancer [34,35,36]. A systematic review by Pan et al. [36] analyzed 39 studies to assess the prognostic value of SUV for survival in patients with esophageal cancer. It has been found that pretreatment SUV measurements can serve as prognostic survival markers in this patient population. However, the SUV threshold was chosen arbitrarily between patients with high and low survival based on the median SUV values in the majority of the studies. Additionally, some studies also used maximum SUV values, reflecting a possible bias. Further, several studies obtained SUV measurements at metastasis sites, rather than the primary tumor, reflecting another difference to our study. Li et al. [34] showed that metabolic parameters of sequential ^18^F-FDG-PET/CTs predict overall survival in esophageal cancer patients treated with chemo-radiation. MVA (which was performed in the aforementioned study), however, revealed that metabolic tumor volume was the only independent prognostic parameter from the initial staging ^18^F-FDG-PET/CT, whereas SUVmax was not found to be significant, which is in line with the results of our study. This may lead to the notion that besides SUV values, additional more-advanced metabolic markers, such as metabolic tumor volume or total lesion glycolysis, should be included as surrogate parameters for outcome prediction.

The following study limitations must be acknowledged. Firstly, there are inherent drawbacks due to the retrospective nature of the study. Secondly, we included patients with both squamous cell carcinoma and adenocarcinoma, leading to relative inhomogeneity of the study cohort. Thirdly, sarcopenia measurements were not performed on post-treatment imaging, since ^18^F-FDG-PET/CT is not funded for restaging purposes in the healthcare system where our study was conducted.

## 5. Conclusions

In conclusion, our study indicates that sarcopenia derived from standard-of-care clinical ^18^F-FDG-PET/CTs is a prognostic marker of poor outcomes in patients with advanced metastatic esophageal and gastroesophageal cancer. Combining the patients’ nutritional states with clinical variables—but not with metabolic activity parameters from ^18^F-FDG-PET/CT—resulted in overall improved prognostic ability regarding OS and PFS.

## Figures and Tables

**Figure 1 diagnostics-13-00838-f001:**
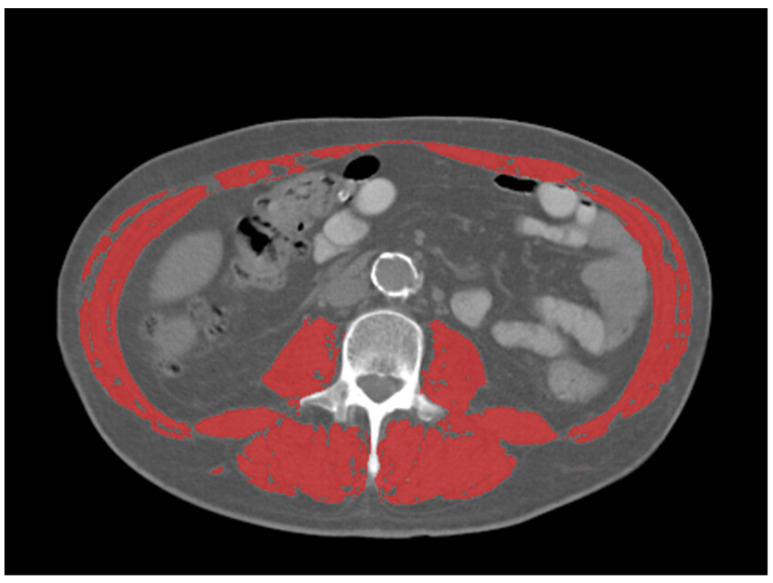
Representative example of skeletal muscle measurements on CT at the level of L3 (outlined in red). Thresholds for skeletal muscle were −29–150 HU.

**Figure 2 diagnostics-13-00838-f002:**
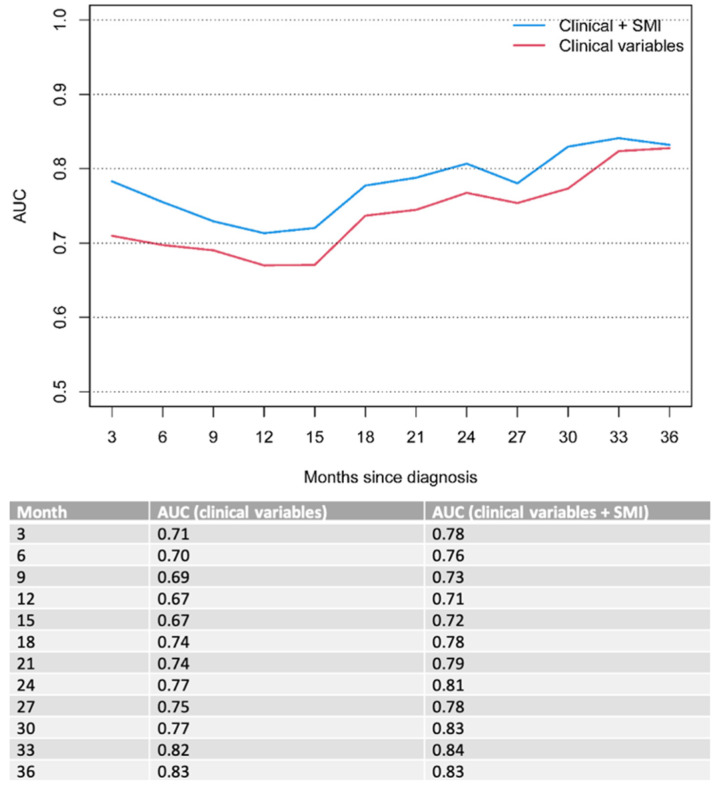
Time-dependent AUC for OS. Clinical variables include ECOG performance status and bone metastases.

**Figure 3 diagnostics-13-00838-f003:**
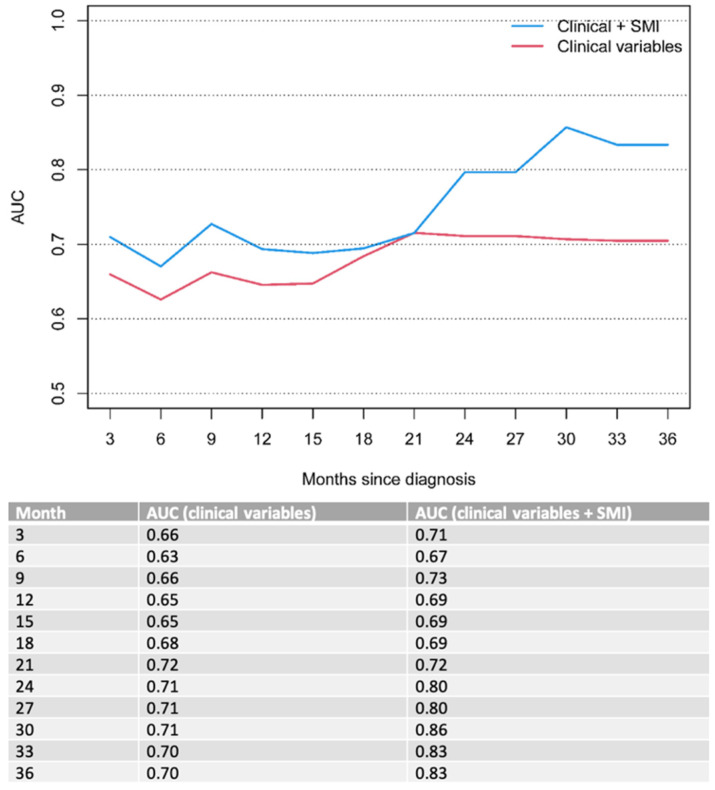
Time-dependent AUC for PFS. Clinical variables include ECOG performance status and bone metastases.

**Table 1 diagnostics-13-00838-t001:** Demographic data of the study cohort.

Characteristics	*n* = 128
Age (mean ± SD; range)	63.5 ± 11.7 (29–91)
Sex	
Females	26 (20%)
Male	102 (80%)
BMI (kg/m^2^) (mean ± SD)	24.4 ± 4.9
Race	
Asian	11 (9%)
Non-Asian	117 (91%)
Siewert Class	
AEG 1: 35–39 cm	27 (21%)
AEG 2: 39–42 cm	27 (21%)
AEG 3: 42–45 cm	15 (12%)
Esophagus: <35 cm	59 (46%)
ECOG	
0	28 (22%)
1	73 (57%)
≥2	27 (21%)
Tumor Grade	
G1-2	47 (37%)
G3	51 (40%)
GX	30 (23%)
T stage	
T0-3	37 (29%)
T4	8 (6%)
TX	83 (65%)
N stage	
N0	6 (5%)
N1	113 (88%)
N2	4 (3%)
NX	5 (4%)
M stage	128 (100%)
Distant Lymph Nodes	73 (57%)
Lung/Pleura	24 (19%)
Liver	43 (34%)
Peritoneum	16 (12%)
Bone	29 (23%)
Brain	2 (2%)
Sarcopenia	60 (47%; 82% males, 18% females)

**Table 2 diagnostics-13-00838-t002:** SUV parameters of the primary tumor.

Metabolic Parameters	Mean (Range)
SUVmax	15.4 (4.1–54.4)
SUVmean	8.4 (2.9–25.2)
SUVpeak	12.9 (3.3–45.7)
SULmax	11.5 (2.6–39.7)
SULmean	6.3 (1.9–18.4)
SULpeak	9.7 (2.3–33.3)

SUV: Standardized Uptake Value.

**Table 3 diagnostics-13-00838-t003:** Univariable Cox regression analysis for OS and PFS in the overall cohort.

Covariate	OS		PFS	
	HR (95%CI)	*p*-value	HR (95%CI)	*p*-value
Age	1.02 (1.00, 1.04)	**0.017**	1.01 (1.00, 1.03)	0.14
Sex (male)	0.90 (0.57, 1.43)	0.65	1.00 (0.64, 1.57)	0.99
Race (non-Asian)	0.69 (0.36, 1.34)	0.28	0.53 (0.27, 1.02)	0.058
ECOG		<**0.001**		<**0.001**
0–1	Reference		Reference	
2–3	3.13 (1.96, 4.98)		2.30 (1.46, 3.62)	
T stage		0.47		0.27
T0-3	Reference		Reference	
T4	1.05 (0.44, 2.55)		0.70 (0.29, 1.68)	0.42
TX	1.30 (0.84, 2.02)		1.25 (0.83, 1.90)	0.28
Tumor Histology		0.68		0.69
Adenocarcinoma	Reference		Reference	
Squamous cell carcinoma	0.92 (0.61, 1.37)		1.08 (0.74, 1.59)	
Tumor Grade		0.77		0.86
G1-2	Reference		Reference	
G3	0.92 (0.60, 1.41)		0.91 (0.60, 1.37)	0.64
GX	1.11 (0.67, 1.85)		1.02 (0.63, 1.66)	0.93
M		0.44		0.45
M1	Reference		Reference	
M1a	0.70 (0.27, 1.79)		0.69 (0.29, 1.65)	0.4
M1b	1.18 (0.77, 1.79)		1.14 (0.76, 1.70)	0.53
Distant LN	0.79 (0.54, 1.16)	0.23	0.91 (0.63, 1.31)	0.61
Lung/ Pleura	1.06 (0.65, 1.73)	0.8	1.09 (0.68, 1.73)	0.73
Liver	1.27 (0.85, 1.89)	0.25	1.15 (0.78, 1.70)	0.47
Peritoneum	1.34 (0.78, 2.32)	0.29	1.01 (0.58, 1.73)	0.98
Bone	1.67 (1.06, 2.63)	**0.028**	1.61 (1.03, 2.51)	**0.038**
Brain	0.49 (0.07, 3.53)	0.48	1.63 (0.40, 6.63)	0.5
SUVmax	0.99 (0.96, 1.01)	0.33	1.00 (0.97, 1.02)	0.86
SUVmean	0.96 (0.91, 1.01)	0.15	0.99 (0.94, 1.04)	0.6
SUVpeak	0.98 (0.95, 1.01)	0.26	0.99 (0.96, 1.02)	0.68
SULmax	0.99 (0.96, 1.02)	0.5	1.00 (0.97, 1.04)	0.83
SULmean	0.96 (0.89, 1.03)	0.23	0.99 (0.92, 1.07)	0.86
SULpeak	0.98 (0.94, 1.02)	0.4	1.00 (0.96, 1.04)	0.99
BMI (kg/m^2^)	0.97 (0.93, 1.02)	0.21	0.98 (0.94, 1.02)	0.3
SMI (cm^2^/m^2^)	0.97 (0.95, 0.99)	**0.0075**	0.97 (0.96, 0.99)	**0.011**
Sarcopenia (yes)	1.51 (1.03, 2.22)	**0.033**	1.55 (1.07, 2.25)	**0.021**

**Table 4 diagnostics-13-00838-t004:** Multivariable Cox regression analysis for OS and PFS in the overall cohort.

Covariate	OS		PFS	
	HR (95%CI)	*p*-Value	HR (95%CI)	*p*-Value
Age	1.01 (0.99, 1.03)	0.2		
ECOG		<**0.001**		<**0.001**
0–1	reference		reference	
2–3	2.91 (1.75, 4.82)		2.27 (1.43, 3.61)	
Bone		**0.01**		**0.019**
No	reference		reference	
Yes	1.84 (1.16, 2.94)		1.71 (1.09, 2.69)	
SMI (cm^2^/m^2^)	0.98 (0.96, 1.00)	0.065	0.98 (0.96, 1.00)	**0.03**

## Data Availability

The data presented in this study are available on request from the corresponding author.

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
