# Peer review of "Prognostic Value of Sarcopenia and Metabolic Parameters of 18F-FDG-PET/CT in Patients with Advanced Gastroesophageal Cancer"

_diagnostics, 2023, doi:10.3390/diagnostics13050838_

Round 1
Reviewer 1 Report
This study indicates that sarcopenia derived from standard of care clinical 18F-FDG-PET/CT is a prognostic marker of poor outcome in patients with advanced, metastatic esophageal and gastroesophageal cancer. Combining the patient’s nutritional state with clinical variables, but not with metabolic activity parameters from 18F-FDG PET/CT, resulted in overall improved prognostic ability regarding OS and PFS. This research topic is relatively new, and the references are normative. The 0.028 of 12th line in the abstract was a writing error.
Author Response
This study indicates that sarcopenia derived from standard of care clinical 18F-FDG-PET/CT is a prognostic marker of poor outcome in patients with advanced, metastatic esophageal and gastroesophageal cancer. Combining the patient’s nutritional state with clinical variables, but not with metabolic activity parameters from 18F-FDG PET/CT, resulted in overall improved prognostic ability regarding OS and PFS. This research topic is relatively new, and the references are normative.
- The 0.028 of 12th line in the abstract was a writing error.
Thank you for sending your respectful feedback. We have corrected the typing error in line 12 of the abstract.
- The English is very difficult to understand/incomprehensive.
We have performed extensive language revision by two native English-speaking colleagues. We hope that the current English level of the manuscript is suitable for publication.
Reviewer 2 Report
The article evaluates baseline staging PET/CTs in patients with advanced, metastatic esophageal and gastroesophageal cancer with a focus on the prognostic value of sarcopenia measurements. The authors found that sarcopenia, ECOG status, and bone metastases were significant prognostic factors for overall survival and progression free survival. Contrary to other published studies, SUV and SUL valves from baseline PET/CTs were not significantly associated with OS and PFS.
Specific comments:
Both the abstract and the discussion mention, “The final model demonstrates improved OS and PFS prognostication when combining clinical parameters with imaging-derived sarcopenia measurements but not metabolic tumor parameters.” However, in the results section, only the univariate analysis of baseline metabolic parameters is discussed. Explicit results from a combination of clinical parameters and metabolic parameters are not described in the methods text. This should be added. I also suggest that the sentence in the abstract be clarified in more detail as to which “metabolic parameters” were evaluated and at which time points (baseline).
The preprocessing of the data including radiomic features removal could be elaborated upon to be made more clear to the reader.
The discussion section describes several prior studies on sarcopenia and gastroesophageal cancer, but does not describe why the current study is novel/original.
The paper is overall very well written with only a few typos (e.g. “clinical parameter” instead of “clinical parameters” in the last sentence of the abstract, “bed potions” instead of “bed positions,” etc.)
The sentence structure to define UVA in section 3.3 is well done and clear. UVA is also defined in section 2.3 but does not actually say “univariable analysis.” This should be fixed because abbreviations only need to be defined once.
Author Response
The article evaluates baseline staging PET/CTs in patients with advanced, metastatic esophageal and gastroesophageal cancer with a focus on the prognostic value of sarcopenia measurements. The authors found that sarcopenia, ECOG status, and bone metastases were significant prognostic factors for overall survival and progression free survival. Contrary to other published studies, SUV and SUL valves from baseline PET/CTs were not significantly associated with OS and PFS.
- Both the abstract and the discussion mention, “The final model demonstrates improved OS and PFS prognostication when combiningclinical parameters with imaging-derived sarcopenia measurements but not metabolic tumor parameters.” However, in the results section, only the univariate analysis of baseline metabolic parameters is discussed. Explicit results from a combination of clinical parameters and metabolic parameters are not described in the methods text. This should be added. I also suggest that the sentence in the abstract be clarified in more detail as to which “metabolic parameters” were evaluated and at which time points (baseline).
Thanks for your suggestion.
To clarify, when we developed the multivariable model, only parameters that are statistically significant in the univariable analyses were selected. The baseline SUV/SUL parameters were not statistically significant in the univariable analyses and thus were not further evaluated in the multivariable analyses. As suggested, we have clarified the sentence in the abstract (p.1).
- The preprocessing of the data including radiomic features removal could be elaborated upon to be made more clear to the reader.
Thank you for your suggestion. We removed two types of features, (i) those with more than 30% missing data, and (ii) those with zero variance, i.e. features that have the same value in all observations. We have modified the wording in the text to make it more clear (p. 4).
- The discussion section describes several prior studies on sarcopenia and gastroesophageal cancer, but does not describe why the current study is novel/original.
Thank you for the respectful feedback. We have added a paragraph to the discussion (p. 9)
- The paper is overall very well written with only a few typos (e.g. “clinical parameter” instead of “clinical parameters” in the last sentence of the abstract, “bed potions” instead of “bed positions,” etc.)
The sentence structure to define UVA in section 3.3 is well done and clear. UVA is also defined in section 2.3 but does not actually say “univariable analysis.” This should be fixed because abbreviations only need to be defined once.
We thank the reviewer for the feedback and have corrected all typing errors and abbreviations.
Thank you again for your efforts. We hope that the current status of our manuscript is suited for publication in Diagnostics.